# Global prevalence, mortality, and main characteristics of HIV-associated pneumocystosis: A systematic review and meta-analysis

Ehsan Ahmadpour[1,2], Sevda Valilou[3], Mohammad Ali Ghanizadegan[4], Rouhollah Seyfi[1], Seyed Abdollah Hosseini[5], Kareem Hatam-Nahavandi[6], Hanieh Hosseini[3], Mahsa Behravan[3], Aleksandra Barac[7], Hamid Morovati[2,8]*

1 Infectious and Tropical Disease Research Center, Tabriz University of Medical Sciences, Tabriz, Iran, 2 Department of Medical Mycology and Parasitology, School of Medicine, Tabriz University of Medical Sciences, Tabriz, Iran, 3 Student Research Committee, Tabriz University of Medical Sciences, Tabriz, Iran, 4 Drug Applied Research Center, Tabriz University of Medical Sciences, Tabriz, Iran, 5 Department of Parasitology and Mycology, Faculty of Medicine, Mazandaran University of Medical Sciences, Sari, Iran, 6 Iranshahr University of Medical Sciences, Iranshahr, Iran, 7 Clinic for Infectious and Tropical Diseases, University Clinical Center of Serbia, Belgrade, Serbia, 8 Department of Parasitology and Mycology, School of Medicine, Shiraz University of Medical Sciences, Shiraz, Iran

* morovatihamid1989@gmail.com

## Abstract

The epidemiology of Human Immunodeficiency Virus (HIV)-associated pneumocystosis (HAP) is poorly described on a worldwide scale. We searched related databases between January 2000 and December 2022 for studies reporting HAP. Meta-analysis was performed using StatsDirect (version 2.7.9) and STATA (version 17) according to the random-effects model for DerSimonian and Laird method and *metan* and *metaprop* commands, respectively. Twenty-nine studies with 38554 HIV-positive, 79893 HIV-negative, and 4044 HAP populations were included. The pooled prevalence of HAP was 35.4% (95% CI 23.8 to 47.9). In contrast, the pooled prevalence of PCP among HIV-negative patients was 10.16% (95% CI 2 to 25.3). HIV-positive patients are almost 12 times more susceptible to PCP than the HIV-negative population (OR: 11.710; 95% CI: 5.420 to 25.297). The mortality among HAP patients was 52% higher than non-PCP patients (OR 1.522; 95% CI 0.959 to 2.416). HIV-positive men had a 7% higher chance rate for PCP than women (OR 1.073; 95% CI 0.674 to 1.706). Prophylactic (OR: 6.191; 95% CI: 0.945 to 40.545) and antiretroviral therapy (OR 3.356; 95% CI 0.785 to 14.349) were used in HAP patients six and three times more than HIV-positive PCP-negatives, respectively. The control and management strategies should revise and updated by health policy-makers on a worldwide scale. Finally, for better management and understanding of the epidemiology and characteristics of this coinfection, designing further studies is recommended.

**Data Availability Statement:** All relevant data are within the manuscript and its Supporting information files.

**Funding:** The author(s) received no specific funding for this work.

**Competing interests:** The authors have declared that no competing interests exist.

## Introduction

Pneumocystis pneumonia (PCP) is a serious pulmonary infection caused by a ubiquitous opportunistic fungal pathogen, *Pneumocystis jirovecii* [1]. Humans are the main reservoir of *P jirovecii*, and inhalation of air-borne particles is the main transmission route [2]. PCP is extremely rare in healthy people, and *P.jirovecii* can live in the lungs without causing symptoms [3]. Exposure to this pathogen is very high [4]. More than 80% of children in developed countries commonly show antibodies against *P.jirovecii* [5, 6]. Up to 20% of adults might carry this fungus at any given time, although the immune system removes the fungus after several months [7]. In 1980 PCP was introduced as a substantial public health problem [8].

The prevalence of PCP escalated following the Human Immunodeficiency Virus (HIV) epidemic in the 1980swhen HIV was diagnosed in a group of homosexual men, hemophilia patients, and recipients of blood products. Although, genetic analysis revealed that the virus was first transmitted to humans around 1908 from chimpanzees polluted SIV virus, probably through the wounds of African people who slaughtered chimpanzees for food. The emergence and increase of the risk factors between 1908 and 1980 led to the HIV epidemic in 1980 [9]. PCP is almost exclusively seen in individuals with compromised immune systems by HIV/ (Acquired Immunodeficiency Syndrome) AIDS. In addition, other immunocompromising statuses include cancer chemotherapy, organ transplantation, long-term corticosteroid therapy, chronic lung diseases, and inflammatory or autoimmune diseases (for example, lupus or rheumatoid arthritis) from PCP leading factors [4, 10, 11]. Also, it causes life-threatening pneumonic disorders in neonates [12]. HIV/AIDS is the main risk factor for 30–40% of PCP cases around the world [13]. PCP is usually a terminal event in HIV/AIDS patients, and much of the information we have about this infection comes from caring for patients with HIV/ AIDS [14].

Recognition of at-risk patients is based on the severity of clinical symptoms(fever, cough, difficulty breathing, chest pain, and tiredness) in high-risk populations [15]. However, radiological examination and specific polymerase chain reaction (PCR) assay in sputum, bronchoalveolar lavage (BAL), tissue biopsies, and serum β-D-glucan levels (a part of the cell wall of many different types of fungi)should support the diagnosis of PCP [16–18]. Liu *et al.* [19] investigated the technical advantages of molecular methods, especially PCR-based methods, for the detection of *P. jirovecii* among BAL fluid specimens.

Without treatment, PCP can lead to severe uncontrolled consequences in the health status [8]. Trimethoprim/sulfamethoxazole(TMP/SMX), also known as co-trimoxazole, is used for PCP treatment and prophylaxis in HIV and non-HIV populations [20–22]. Recently, progressive achievements in designing antiretrovirals have decreased the prevalence of HIV-associated PCP (HAP) [23]. The mortality rate of PCP is elevated due to the abrupt occurrence of respiratory failure and delay in diagnosis [24, 25].

Here, we designed this analysis to provide accurate statistics of AIDS-defining PCP. The results of our study will be suitable for researchers and health policy-makers to manage and develop preventative strategies for infection control.

## Methods

### Search strategy

The present study is conducted and reported according to PRISMA 2020 guideline [26] (S1 Checklist). We developed a broad search strategy to identify studies that reported PCP among HIV/AIDS population. In our systematic review, the search terms "HIV/AIDS,""HIV,""AIDS,""*Pneumocystis*,""*Pneumocystis jirovecii*,""pneumocystosis,""Human

Immunodeficiency Virus,"and related terms and words for relevant studies published in PubMed, Web of Science, Scopus, Google Scholar, and ProQuest between January 2000and December 2022 were used. No linguistic or geographical limits were applied. We hand-searched bibliographies of all recovered articles for potentially eligible studies and contacted corresponding authors for published or unpublished data if needed. January 2000 was chosen as the cut-off because we aimed to assay studies published during the past 22 years.

## Selection criteria

Titles and abstracts of references were screened, and the full texts of potentially relevant articles were independently assessed. Studies assessing a clearly defined population of HIV/AIDS and PCP in any clinical setting were included if they had specific diagnostic criteria for PCP. These were predefined using clinical case definitions [27] or confirmation with laboratory testing using molecular assays, such as PCR, sequencing, and matrix-assisted laser desorption-ionization time of flight mass spectrometry (MALDI-TOF MS) [16–18]. Inclusion criteria were as follows; patients with HIV/AIDS and PCP, all types of studies encompassing data about patients with HIV/AIDS and PCP infected simultaneously, including clinical trials, retrospective, prospective, and cohort studies, gray literature including conference reports, etc. Exclusion criteria were as follows; patients with HIV/AIDS and without PCP or patients who have other fungal infections than pneumocystosis, all review type studies (*e.g.*, narrative, critical, systematic, and meta-analysis, and mini-reviews) case reports and case series, all studies including letters to the editor, and editorials, without patient data.

## Data extraction

Three authors independently extracted data and compared it for consistency. Discussion and consensus resolved disagreements on final inclusions. The key variable was the proportion of PCP coinfection among the HIV/AIDS population. Prevalence was defined as the number of PCP cases (nominator) among patients with established HIV/AIDS(denominator). The following information was captured where available; PCP among HIV-negative patients, underlying risk factors, PCP prophylaxis, antiretroviral therapy, age and gender of the target population, laboratory parameters (*e.g.*, serum levels of CD4$^+$ T cells, β-D-glucan, and lactate dehydrogenase), and the outcome of patients (death or survival).

## Risk of bias (quality) assessment

This research involved studies concerning a minimum of three participants to minimize the small-study effect. Authors independently assessed the quality according to the Hoy *et al.* checklist previously described [28, 29]. This checklist explored the various dimensions of empirical proof and methodological assumptions. If required, a consensus was voted by other coauthors to settle the disputes between the investigators. Moreover, the regression-based Egger, Begg's-Mazumdar, and Harbord tests for small-study effects will apply to analyze publication bias for our search [30].

## Data analysis

Meta-analysis was performed according to the DerSimonian and Laird method [31, 32], applying the random-effects model [33] in case of considerable heterogeneity, defined as $I^2 > 75\%$. We evaluated heterogeneity using the chi-square ($\chi^2$-based Q statistic, significant for *P*value0<0.5) and the $I^2$ statistic. StatsDirect version 2.7.9 (StatsDirect Ltd, Wirral-UK)and STATA version 17 (STATA corporation, USA) were used to perform calculations and the

meta-analysis [34, 35]. Analysis via STATA was performed using the *metan* and *metaprop* commands [36, 37] for standard mean difference (SMD) calculation. Odds ratio (OR) analysis was performed for related data if their case(s) and control(s) details were available. Point estimates and 95% confidence intervals were derived using prevalence data from included studies for all outcomes. Where standard errors (SE) were not provided, we incorporated confidence intervals into the formula, SE = (upper limit–lower limit)/3.92. Subgroup analysisand meta-regression were used to determine the source of heterogeneity based on certain putative moderator factors, and sensitivity analysis was used to assess the reliability of our pooling results.

## Results

After searches in the databases and removing duplicate and irrelevant records, our meta-analysis included fifty-five eligible studies (_((((xxx))))_)[38–92] (Table 1; Fig 1). The results of the quality assessment were added to Table 1. The overall score of 4.378 was reached for quality assessment, which means analyzed studies had a moderate risk of bias (Table 1). A total of38,554HIV/AIDS patients were included, and PCP was found in 2,880of them. Also, 4,044 PCP cases were reported among 79,893 HIV/AIDS-negative patients.

Five studies were conducted from the USA [50–52, 68, 73]; three studies, each were conducted in South Africa [67, 72, 90], Japan [81, 84, 85], and Thailand [61, 78, 86]; two studies, each were conducted in Iran [75, 80], France [45, 66], and Mexico [40, 47]; one study each was conducted in Ethiopia [39], Tanzania [60], Malawi [57], Spain [38], Cameroon [76], India [54], Taiwan [89], Zambia [79], and a study was from five European countries (France, Austria, Belgium, Croatia, Germany) [70].

Almost all analyzed studies targeted respiratory tract samples to detect PCP among HIV/AIDS patients. In some cases, serum specimen was targeted. Different diagnostic methods, *e. g.*, conventional and real-time PCR, targeting *mitochondrial small and large subunits* (*mtSSU* and *mtLSU*), fast-track diagnostic (FTD) methods capturing β-D-glucan, and microscopical diagnosis were applied.

### The pooled prevalence of HIV-associated PCP

Our random-effects model showed that in twenty-nine eligible studies [38–40, 45, 47, 50–52, 54, 57, 60, 61, 66–68, 70, 72, 73, 75, 76, 78–81, 84–86, 89, 90], the pooled prevalence of HAP was 35.4% (95% CI:23.8 to 47.9; $I^2$: 99.4%) (Table 2). There is a negligible publication bias between studies (intercept: 6.946; 95% CI:-24.750 to 38.642; P = 0.6566). The percent rates of HAP by country in twenty-nine eligible studies were as follows: USA5.16% (1854/35885), South Africa22.8% (128/559), Thailand64.66% (97/150), Japan 97.79% (443/453), Iran 12.35% (21/170), France 27.93% (69/247), Mexico 42.27%(52/110), Ethiopia 22.79% (49/215), Tanzania 0.469% (1/213), 11.19% (15/134), Spain 37.5% (3/8), Cameroon 42.85% (54/126), Taiwan 82.35% (14/17), Zambia 25% (47/188), India 33.33%(22/66), and a study in the five European countries 84.61% (11/13).

### The pooled prevalence of PCP among the HIV-negative population

The results of the random-effect model showed that in twenty-nine eligible studies [38–40, 45, 47, 50–52, 54, 57, 60, 61, 66–68, 70, 72, 73, 75, 76, 78–81, 84–86, 89, 90], the pooled prevalence of PCP among the HIV-negative population was 10.158% (95% CI:2 to 25.3; $I^2$: 99.9%). Also, analysis of publication bias using the Egger test model indicated an intercept rate of 14.326% (95% CI: -2.023 to 30.675; P = 0.0834)(Table 2). By country percent rates are as follows: USA 0% (0/70418), Zambia 0% (0/1115), Tanzania 0% (0/279), South Africa 4.40% (25/567), Thailand 9.83% (36/366), Japan 84.73% (3631/4285), Iran 6.52% (21/322), Ethiopia 3.55% (6/169),

**Table 1. Notice: Some of the studies in this table were not included in the odds-ratio analysis due to heterogeneity and bias assessment problems [110–117].** However, they were included in other analyses. YOP: year of publish, HIV: human immunodeficiency virus, PCP: pneumocystosis, HAP: HIV-associated pneumocystis, -: negative, +: positive, LDH: lactate dehydrogenase, BDG: β-D-glucan, ART: antiretroviral therapy, PFX: prophylaxis, QA: quality assessment, ND: not defined, L: low, M: moderate, H: high.

| First author; YOP; Country | Study Design | HIV positive | HAP | HIV negative | HIV-PCP + | HAP Gender | HAP Mean Age | HAP Mortality | HAP Mean CD4 | HAP Mean LDH | HAP Mean BDG | HAP ART | HAP PFX | QA Score; Status |
|---|---|---|---|---|---|---|---|---|---|---|---|---|---|---|
| Park et al.; 26 June 2001; USA | Single-center prospective | 112 | 35 | 410 | 0 | ND | ND | ND | ND | ND | ND | ND | 35 | 5; M |
| Daly et al.; 9 Aug 2002; USA | Cross-sectional | 94 | 34 | 95 | 0 | ND | ND | ND | ND | ND | ND | ND | ND | 5; M |
| Aderaye et al.; 21 Oct 2002; Ethiopia | Retrospective | 215 | 49 | 169 | 6 | ND | ND | ND | ND | ND | ND | ND | ND | 7; H |
| Zar et al.; 2 Jan 2007; South Africa | Cohort | 151 | 15 | 99 | 4 | ND | ND | 7 | ND | ND | ND | ND | 15 | 2; L |
| Tasaka et al.; 1 April 2007; Japan | Cross-sectional | 16 | 13 | 263 | 44 | ND | ND | ND | ND | ND | ND | ND | ND | 6; M |
| Jensen et al.; 28 May 2010; Tanzania | Prospective | 205 | 1 | 279 | 0 | Men: 1 W: 0 | ND | 1 | 210 | ND | ND | ND | 1 | 3; L |
| Crothers et al.; 29 Dec 2010; USA | Cross-sectional | 153 | 3 | 92 | 0 | ND | ND | ND | ND | ND | ND | ND | ND | 7; H |
| Crothers et al.; 1 Feb 2011; USA | Cohort | 33420 | 1771 | 66840 | 0 | ND | ND | ND | ND | ND | ND | 202 | ND | 2; L |
| Graham et al.; 8 Jun 2011; Malawi | Prospective | 134 | 15 | 130 | 4 | ND | ND | 11 | ND | ND | ND | 11 | 15 | 1; L |
| Acosta et al.; July 2011; Spain | Prospective | 8 | 3 | 43 | 1 | ND | ND | ND | ND | ND | ND | ND | ND | 6; M |
| Tia et al.; 31 Aug 2011; Thailand | Cross-sectional | 66 | 47 | 36 | 7 | ND | ND | ND | ND | ND | ND | ND | ND | 7; H |
| Morrow et al.; 10 Jan 2014; South Africa | Prospective | 124 | 87 | 76 | 21 | ND | ND | ND | ND | ND | ND | ND | ND | 3; L |
| Riebold et al.; 19 Mar 2014; Cameroon | Single-center prospective | 126 | 54 | 111 | 21 | Men: 8 W: 46 | ND | ND | ND | ND | ND | 44 | ND | 0; L |
| Tasaka et al.; 24 July 2014; Japan | Cross-sectional | 23 | 16 | 143 | 48 | ND | ND | ND | ND | ND | ND | ND | ND | 5; M |
| Sheikholeslami et al.; 28 Jan 2015; Iran | Cross-sectional | 130 | 16 | 242 | 18 | ND | ND | ND | ND | ND | ND | ND | ND | 4; M |
| Bienvenu et al.; 18 March 2016; France | Retrospective | 143 | 143 | ND | ND | ND | 59 | ND | ND | ND | ND | ND | ND | 2; L |
| Louis et al.; 18 Nov 2018; France | Retrospective | 180 | 52 | 823 | 27 | ND | ND | ND | ND | ND | ND | ND | ND | 3; L |
| Alcántara-Mojica et al.; Feb 2019; Mexico | Cross-sectional prospective | 43 | 11 | 4 | 0 | ND | 36.2 | ND | ND | ND | ND | ND | ND | 3; L |
| Marjani et al.; Feb 2019; Iran | Retrospective | 710 | 26 | ND | ND | ND | ND | ND | ND | ND | ND | 26 | ND | 3; L |
| Christe et al.; 10 March 2019; Switzerland | Multi-center retrospective | 60 | 60 | ND | ND | ND | 43 | ND | 79 | ND | ND | ND | ND | 7; H |
| Almaghrabi et al.; March 2019; Saudi Arabia | Single-center retrospective | 20 | 20 | ND | ND | ND | 42 | ND | ND | ND | ND | ND | ND | 4; M |
| Azimi et al.; April 2019; Iran | Cross-sectional | 102 | 3 | ND | ND | ND | ND | ND | ND | ND | ND | 3 | ND | 3; L |

*(Continued)*

**Table 1.** (Continued)

| First author; YOP; Country | Study Design | HIV positive | HAP | HIV negative | HIV-PCP + | HAP Gender | HAP Mean Age | HAP Mortality | HAP Mean CD4 | HAP Mean LDH | HAP Mean BDG | HAP ART | HAP PFX | QA Score; Status |
|---|---|---|---|---|---|---|---|---|---|---|---|---|---|---|
| Fraczek et al.; 30 April 2019; UK | Cross-sectional | 160 | 4 | ND | ND | ND | 49.2 | ND | 366 | ND | ND | 1 | 1 | 6; M |
| Singh et al.;June 2019; India | Cross-sectional | 76 | 17 | ND | ND | Men: 14 W: 3 | 34.6 | ND | ND | ND | ND | 2 | 15 | 6; M |
| Hammarström et al.; 23 July 2019; Sweden | Retrospective Case-control | 47 | 26 | ND | ND | ND | 47 | ND | 35 | ND | ND | ND | ND | M; 5 |
| Carreto-Binaghi et al.; Sep 2019; Mexico | Single-center prospective | 84 | 41 | 205 | 16 | ND | ND | ND | ND | ND | ND | ND | ND | M; 6 |
| Wójtowicz et al.; 1 Sep 2019; Swiss | Multi-center prospective | 3605 | 500 | ND | ND | ND | ND | ND | ND | ND | ND | ND | ND | L; 3 |
| Yang et al. 7 Oct 2019; Taiwan | Cross-sectional | 17 | 14 | 124 | 107 | ND | ND | ND | ND | ND | ND | ND | ND | M; 6 |
| Figueiredo et al.; Oct 2019; Portugal | Cross-sectional | 75 | 75 | ND | ND | ND | 45.5 | ND | ND | ND | ND | ND | ND | M; 6 |
| Bozorgomid et al.; Nov 2019; Iran | Retrospective | 114 | 26 | ND | ND | ND | ND | ND | ND | ND | ND | ND | 26 | M; 6 |
| Schäfer et al.; 15 Nov 2019; Germany | Multi-center prospective | 61 | 50 | ND | ND | ND | ND | ND | ND | ND | ND | ND | ND | M; 6 |
| Kato et al.; Nov 2019; Japan | Prospective | 31 | 31 | ND | ND | ND | 41 | ND | 338 | 338 | 29 | ND | ND | M; 5 |
| Rilinger et al.; 14 Nov 2019; Germany | Single-center | 6 | 6 | ND | ND | ND | 36.8 | ND | ND | ND | ND | ND | ND | M; 5 |
| Maartens et al.; 15 March 2020; South Africa | Single-center retrospective | 284 | 26 | 392 | 0 | ND | ND | ND | ND | ND | ND | 26 | ND | M; 6 |
| Shoji et al.; 1 Aug 2020; Japan | Retrospective observational | 414 | 414 | 3879 | 3539 | ND | ND | ND | ND | ND | ND | ND | ND | 1; L |
| Zhu et al.; 10 Sep 2020; China | Cross-sectional | 147 | 60 | ND | ND | ND | 43.17 | ND | 21 | ND | ND | ND | ND | M; 6 |
| de Armas et al.; Nov 2020; Cuba | Retrospective observational | 514 | 53 | ND | ND | ND | ND | ND | ND | ND | ND | ND | ND | M; 6 |
| Yanagisawa et al.; 23 Dec 2020; Thailand | Cohort | 632 | 96 | ND | ND | ND | ND | ND | ND | ND | ND | ND | ND | 3; L |
| Mercier et al.; 1 Dec 2020; Belgium, France, Germany, Croatia, Austria | Multi-center cross-sectional | 13 | 11 | 134 | 104 | ND | ND | ND | ND | ND | ND | ND | ND | M; 6 |
| Kim et al.; 4 Feb 2021; Korea | Retrospective | 26 | 26 | ND | ND | ND | 44.5 | 4 | ND | 778 | ND | ND | ND | 3; L |
| Juniper et al.; 26 April 2021; UK | Cross-sectional | 76 | 29 | ND | ND | Men: 21 W: 8 | 48 | ND | 19 | ND | ND | 1 | 29 | 5; M |
| Makinson et al.; 19 Jul 2021; USA | Prospective | 2106 | 11 | 2981 | 0 | ND | ND | ND | ND | ND | ND | ND | ND | 4; M |
| Rafat et al. Aug 2021; Iran | Prospective cross-sectional | 40 | 5 | 80 | 3 | ND | ND | ND | ND | ND | ND | ND | ND | M; 6 |
| Moore et al.; 25 Aug 2021; South Africa | Multi-center prospective | 115 | 38 | ND | ND | ND | ND | ND | ND | ND | ND | 8 | 28 | 3; L |
| Azovtseva et al.; Sep 2021; Russia | Retrospective | 85 | 23 | ND | ND | ND | ND | ND | 37 | ND | ND | ND | ND | M; 6 |

(Continued)

**Table 1.** (Continued)

| First author; YOP; Country | Study Design | HIV positive | HAP | HIV negative | HIV-PCP + | HAP Gender | HAP Mean Age | HAP Mortality | HAP Mean CD4 | HAP Mean LDH | HAP Mean BDG | HAP ART | HAP PFX | QA Score; Status |
|---|---|---|---|---|---|---|---|---|---|---|---|---|---|---|
| Seidenberg et al. Sep 2021; Zambia | Multi-center prospective | 188 | 47 | 1115 | 0 | ND | ND | ND | ND | ND | ND | ND | 36 | 2; L |
| Jitmuang et al.; 22 Sep 2021; Thailand | Prospective cross-sectional | 15 | 8 | 44 | 4 | ND | 39 | 0 | 22 | ND | ND | ND | ND | 3; L |
| Kasahara et al.; 11 Oct 2021; Japan | Retrospective cohort | 133 | 91 | ND | ND | ND | 43.9 | ND | ND | ND | ND | 91 | ND | 3; L |
| Sarasombath et al.; 20 Oct 2021; Thailand | Cross-sectional | 69 | 42 | 286 | 25 | ND | ND | ND | ND | ND | ND | ND | ND | 3; L |
| Dhanalakshmi et al.; 30 Dec 2021; India | Cross-sectional | 66 | 22 | 223 | 37 | ND | ND | 4 | ND | ND | ND | ND | ND | 5; M |
| Huang et al.; 20 Jan 2022; China | Multi-center prospective | 356 | 356 | ND | ND | ND | 47.45 | ND | 22 | ND | 360.9 | ND | ND | 6; M |
| Tang et al.; 28 Jan 2022; China | Retrospective | 70 | 13 | ND | ND | ND | ND | ND | ND | ND | ND | ND | ND | 6; M |
| Chagas et al.; 24 Feb 2022; Brazil | Cross-sectional | 86 | 41 | ND | ND | Men: 33 W: 8 | 36 | ND | ND | 397.7 | ND | ND | 30 | 1; L |
| Pansu et al.; 2 Mar 2022; France | Prospective cohort | 1736 | 19 | ND | ND | ND | ND | ND | ND | ND | ND | ND | ND | 3; L |
| Feng et al.; 15 Mar 2022; China | Retrospective | 193 | 193 | ND | ND | ND | 38.12 | 35 | 88 | 402 | 88 | ND | ND | 3; L |
| Xue et al.; Mar 2022; China | Cross-sectional | 2 | 1 | ND | ND | ND | 58 | 0 | 204 | 565.5 | ND | ND | ND | 6; M |
| Spottiswoode et al.; 25 Mar 2022; Uganda | Retrospective observational | 217 | 12 | ND | ND | ND | ND | ND | ND | ND | ND | ND | 12 | 7; H |
| Qiao et al.; 25 Mar 2022; China | Prospective | 18 | 10 | ND | ND | Men: 5 W: 5 | 46.2 | ND | 126 | ND | ND | ND | ND | 6; M |
| Ali et al.; 20 May 2022; Qatar | Retrospective | 167 | 42 | ND | ND | ND | ND | ND | ND | ND | ND | ND | ND | 3; L |
| Ueckermann et al.; 15 Jun 2022; South Africa | Retrospective cohort | 117 | 25 | ND | ND | ND | ND | 12 | ND | ND | ND | ND | ND | 4; M |
| Bigot et al.; 24 Jun 2022; France | Retrospective | 67 | 17 | 575 | 8 | ND | ND | ND | ND | ND | 523 | ND | ND | 4; M |
| Zeng et al.; 25 Aug 2022; China | Multi-center prospective | 363 | 363 | ND | ND | ND | 47 | 46 | 24 | ND | ND | ND | ND | 3; L |

Malawi 3.07% (4/130), Spain 2.32% (1/43), Cameroon 18.9% (21/111), Taiwan 86.29% (107/124), India 16.59% (37/223), and a study in the five European countries 77.61% (104/134).

## The relationship between HIV/AIDS and susceptibility to PCP

The results of the random effect model in twenty-nine included studies [38–40, 45, 47, 50–52, 54, 57, 60, 61, 66–68, 70, 72, 73, 75, 76, 78–81, 84–86, 89, 90] indicated that the prevalence of PCP among the HIV-positive population is almost 12 times higher than HIV-negative population (OR: 11.710; 95% CI:5.420 to 25.297;$I^2$: 93.1%; P<0.0001) (Table 2; Fig 2). Also, the Egger test assay result indicated a negligible publication bias among analyzed studies (intercept: 1.416; 95% CI: 0.133 to 2.966; P = 0.1437) (Table 2; S1 Fig).

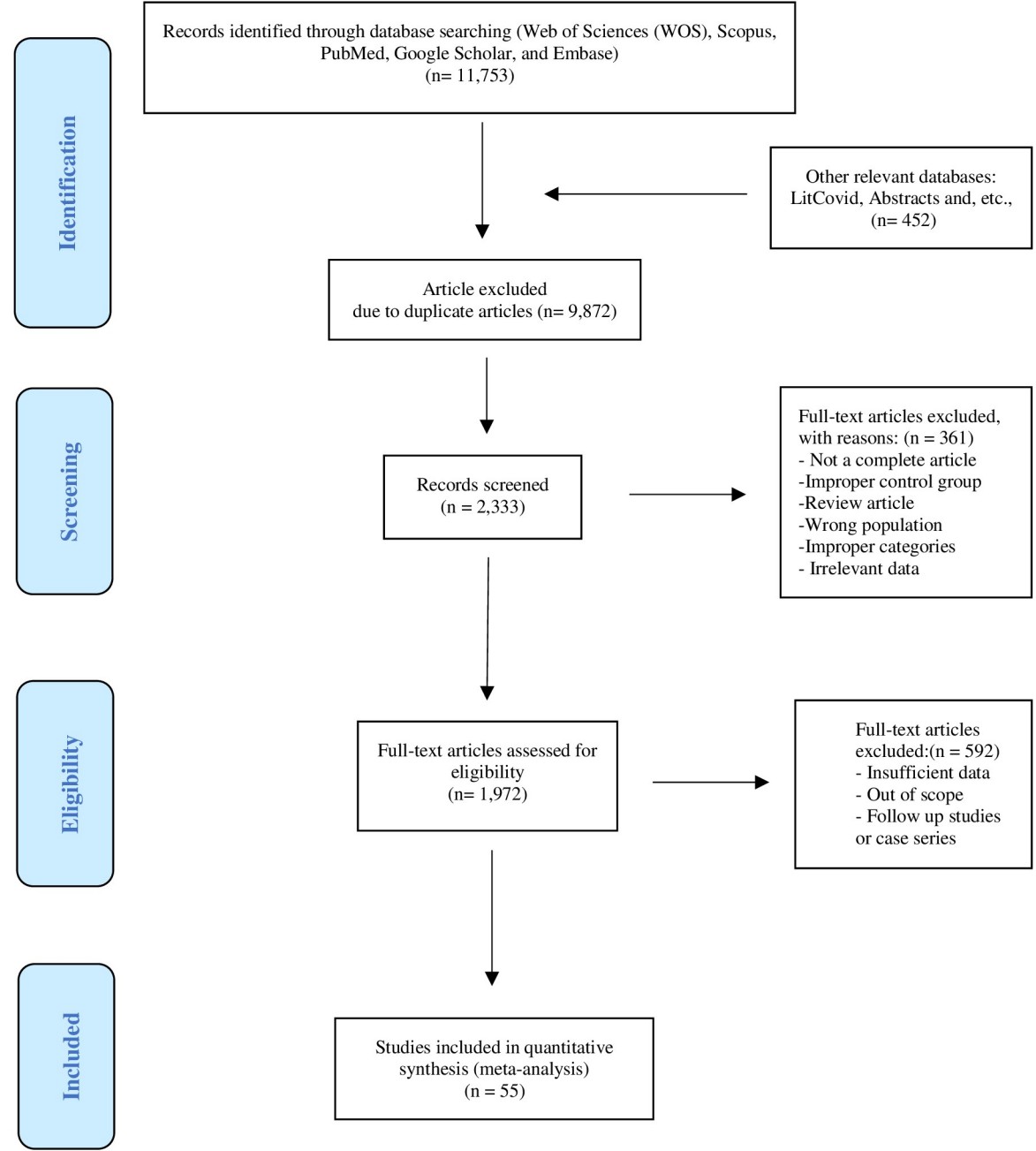

**Fig 1. The flowchart of study identification and selection process.**

## The intercontinental distribution of HIV-associated PCP

Our random-effects model showed that the pooled prevalence of HAP in four continents is as follows: America13% (95% CI:10 to 17; $I^2$: 99.2%), Africa 24%(95% CI: 12 to 35; $I^2$: 98.61%), Asia 52% (95% CI: 32 to 73; $I^2$: 96.11%), and Europe 43% (95% CI: 22 to 64; $I^2$: 90.05%) (Tables 2 and 3).

**Table 2.** HIV: human immunodeficiency virus, PCP: pneumocystosis, PRP: proportion, OR: odds ratio, SMD: standard mean difference, AM: America, AF: Africa, AS: Asia, EU: Europe, M: men, W: women, P: positive, N: negative, LDH: lactate dehydrogenase, BDG: β-D-glucan, ART: antiretroviral therapy, TB: tuberculosis.

| Variables & Risk Factors | # Of studies | # of PCP cases/ HIV cases | Size Effect (%) (95%CI) Random effects (DerSimonian-Laird) | Chi²* P-value | Heterogeneity | | Bias Indicator |
|---|---|---|---|---|---|---|---|
| | | | | | I²(%) (95% CI) | Cochran Q P-value | Egger's test Intercept (95% CI) P-value |
| Prevalence of PCP among HIV positive | 29 | 2880/38554 | PRP: 35.395 (23.8 to 47.9) | - | 99.4 (99.3 to 99.4) | 4485.996 (df = 28) P < 0.0001 | 6.946 (-24.750 to 38.642) P = 0.6566 |
| Prevalence of PCP among HIV negative | 29 | 4044/79893 (HIV negative) | PRP: 10.158 (2.0 to 25.3) | - | 99.9 (99.9 to 99.9) | 25896.639 (df = 28) P < 0.0001 | 14.326 (-2.023 to 30.675) P = 0.0834 |
| OR prevalence | 29 | 2880/38554 | OR:11.710 (5.420 to 25.297) | 39.202 (df = 1) P <0.0001 | 93.1 91.5 to 94.2 | 404.645 (df = 28) P < 0.0001 | 1.416 -0.133 to 2.966 P = 0.1437 |
| AM | 7 | 1906/35995 | PRP: 13 (10 to 17) | 745.45 (df = 6) P = 0.00 | 99.2 (83.6 to 99.9) | ND | 4.183 (-9.654 to 18.020) P = 0.472 |
| AF | 8 | 294/1435 | PRP: 24 (12 to 35) | 502.37 (df = 7) P = 0.00 | 98.61 (87.34 to 99.1) | ND | 9.800 (4.723 to 14.877) P = 0.003 |
| AS | 9 | 597/856 | PRP: 52 (32 to 73) | 205.83 (df = 8) P = 0.00 | 96.11 (85.51 to 98.01) | ND | 8.602 (2.214 to 14.989) P = 0.015 |
| EU | 4 | 83/268 | PRP: 43 (22 to 64) | 30.14 (df = 3) P = 0.00 | 90.05 (81.04 to 93.78) | ND | 3.323 (-10.249 to 16.896) P = 0.403 |
| Mortality | 11 | 696/1298 | OR: 1.522 (0.959 to 2.416) | 19.47 (df = 9) P = 0.022 | 53.8 (47 to 67.5) | 60.866 (df = 10) P <0.0001 | 3.961 (-1.465 to 9.389) P = 0.131 |
| Gender (M to W) | 6 | Women: 70/270 Men: 82/317 | OR: 1.073 (0.674 to 1.706) | 1.04 (df = 5) P = 0.959 | 0 (0 to 0) | 2.204 (df = 6) P = 0.899 | 1.097 (-0.546 to 2.741) P = 0.137 |
| Age | 21 | 1583/2044 | SMD: -0.140 (-0.315 to -0.034) | 93.29 (df = 20) P = 0.000 | 78.6 (71 to 83.2) | 4.349 (df = 11) P <0.001 | -0.871 (-0.613 to -1.796) P = 0.320 |
| CD4 | 12 | 830/1242 | SMD: -0.617 (-1.122 to -0.111) | 223.2 (df = 11) P = 0.000 | 95.1 (83 to 98) | 3.023 (df = 8) P < 0.0001 | -2.769 (-7.650 to 2.11) P = 0.235 |
| LDH | 5 | 293/338 | SMD: 0.089 (-0.089 to 0.267) | 4.36 (df = 4) P = 0.36 | 8.2 (2.235 to 22.560) | 1.875 (df = 4) P < 0.0001 | 0.331 (-2.88 to 3.551) P = 764 |
| BDG | 5 | 284/379 | SMD: 2.664 (1.110 to 4.217) | 175.69 (df = 4) P = 0.000 | 97.7 (53.78 to 99.34) | 2.342 (df = 7) P < 0.0001 | 6.381 -6.286 to 19.049 P = 0.207 |
| ART (OR: P to N) | 11 | HIV⁺ PCP⁺:2277 HIV⁺PCP⁻: 33267 | OR: 3.356 (0.785 to 14.349) | 2.669 (df = 1) P = 0.0012 | 89.7 (83.9to 92.7) | 96.702 (df = 10) P < 0.0001 | 0.051 -2.517 to 2.620 P = 0.9647 |
| PCP prophylaxis (OR: P to N) | 12 | HIV⁺ PCP⁺:368 HIV⁺PCP⁻: 1367 | OR: 6.191 (0.945 to 40.545) | 3.615 (df = 1) P = 0.0472 | 90.4% (85.7 to 93.1) | 114.841 (df = 11) P < 0.0001 | -0.738 (-4.133 to 2.656) P = 0.6384 |

(*Continued*)

**Table 2.** (Continued)

| Variables & Risk Factors | # Of studies | # of PCP cases/ HIV cases | Size Effect (%) (95%CI) Random effects (DerSimonian-Laird) | Chi²* P-value | Heterogeneity | | Bias Indicator |
|---|---|---|---|---|---|---|---|
| | | | | | $I^2$(%) (95% CI) | Cochran Q P-value | Egger's test |
| | | | | | | | Intercept (95% CI) P-value |
| Smoker (OR: P to N) | 3 | HIV⁺ PCP⁺:89 HIV⁺PCP⁻: 78 | OR: 2.250 (0.578 to 8.753) | 1.370 (df = 1) P = 0.242 | 0 (0 to 72.9) | 0.098 (df = 2) P = 0.952 | NA |
| TB (OR: P to N) | 3 | HIV⁺ PCP⁺:355 HIV⁺PCP⁻: 315 | OR: 2.152 (0.311 to 14.879) | 0.604 (df = 1) P = 0.4372 | 92.3 (78.5 to 95.8) | 25.846 (df = 2) P < 0.0001 | NA |

## Relationship between mortality and PCP among HIV-positive patients

In eleventen studies [54, 55, 57, 60, 61, 65, 87, 88, 90, 91], including 670 PCP cases among 1272 HIV-positive patients, death events were recorded in 124 and 210 cases, respectively. The results of our OR analysis indicated that the mortality rate increased by 52% in HAP patients (OR:1.522; 95% CI:0.959 to 2.416;$I^2$: 53.8%; P = 0.022)(Table 2).

## Relationship between PCP prophylaxis and HIV ART and PCP occurrence among HIV-positive patients

Related data were captured from twelve (for PCP prophylaxis) [46, 48, 56, 57, 60, 62, 71, 73, 79, 82, 83, 90] and eleven (for ART) [42, 51, 57, 60, 62, 63, 67, 69, 71, 76, 82] studies. The OR analysis results indicated that PCP prophylaxis in HIV/PCP-positive patients was used six times more than in HIV-positive PCP-negative patients (OR: 6.191; 95% CI:0.945 to 40.545; $I^2$: 90.4%; P = 0.0472)(Table 2). Also, ART in HIV/PCP-positive patients was used three times more than in HIV-positive PCP-negative patients (OR: 3.356; 95% CI: 0.785 to 14.349;$I^2$: 89.7%; P = 0.0012) (Table 2).

## Relationship between age and sex with PCP occurrence among HIV-positive patients

The results of the OR analysis of patient gender captured from six studies [48, 60, 62, 74, 76, 82] indicated that HIV-positive men had a 7% higher chance rate for PCP compared to women (OR:1.073; 95% CI:0.674 to 1.706;$I^2$: 0%; P = 0.959)(Table 2). The results of standard mean difference (SMD) analysis of patient's age captured from twenty-one studies [40, 41, 44, 48, 49, 53, 55, 56, 58, 59, 61–65, 74, 77, 82, 88, 91, 92], including 1583 PCP patients among 2044 HIV-positive cases, indicated that there is no significant difference between HIV/AIDS patients with PCP and without PCP (SMD: -0.140; 95% CI:-0.315 to -0.034;$I^2$: 78.6%; P = 0.000).

## Variation of serological factors and/or BAL levels between HIV-positive patients with and without PCP

Our random effect models for SMD analysis of serum levels of CD4⁺ T cells [43, 49, 55, 56, 58–62, 74, 88, 91, 92], LDH [48, 55, 64, 65, 88], and BDG [45, 55, 59, 64, 92] in 12, 5, and 5 studies (respectively), indicated that serum levels of CD4⁺ T cells (SMD: -0.617; 95% CI:-1.122 to -0.111; $I^2$: 95.1%;P = 0.000) and LDH (SMD: -0.089; 95% CI:-0.089 to 0.267; $I^2$: 8.2%;

## Odds ratio meta-analysis plot [random effects]

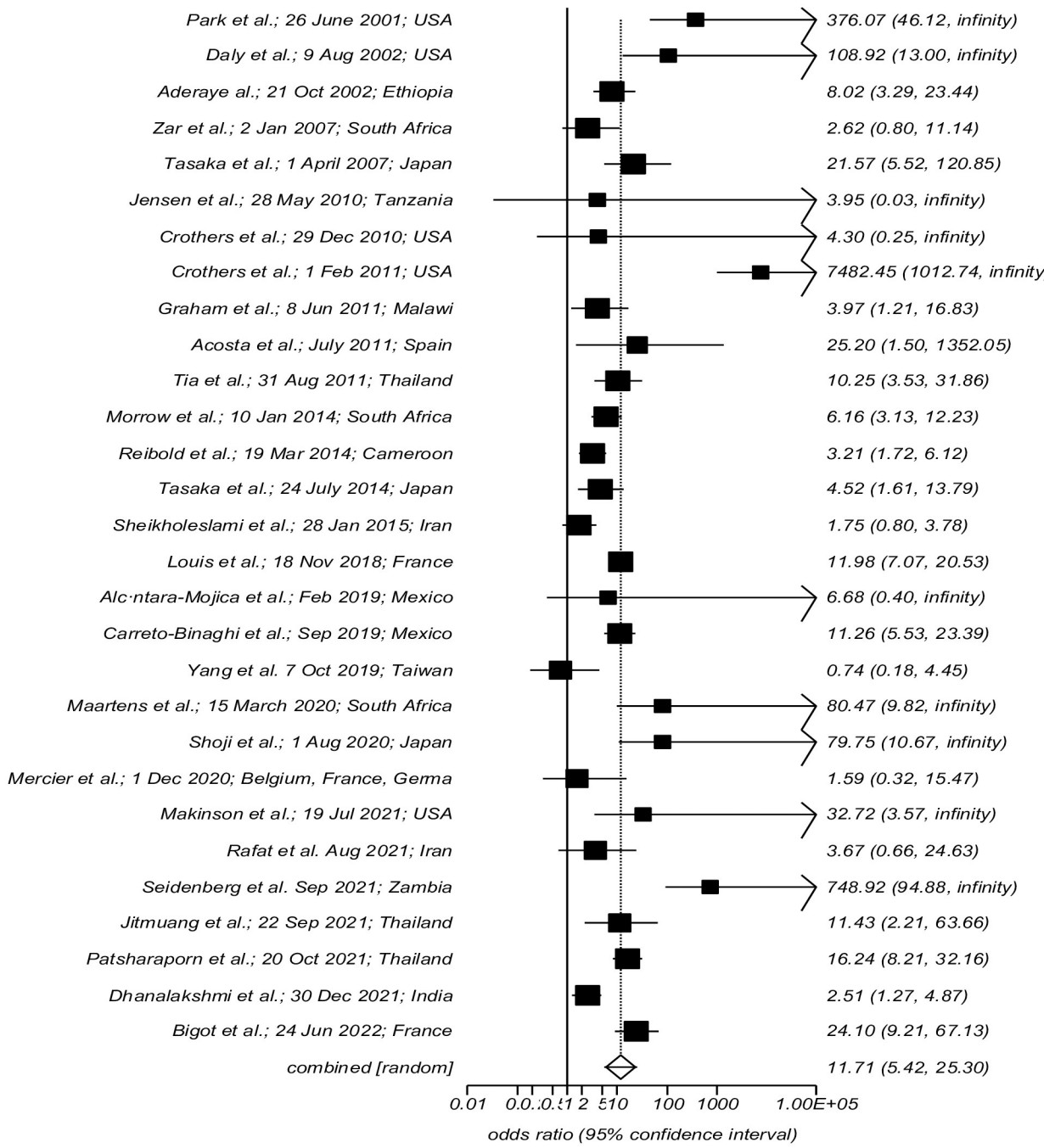

**Fig 2. Forest plot of the prevalence OR of HAP.**

P = 0.36) did not differ statistically between HIV/PCP-positive and HIV-positive PCP-negative patients. However, the mean serum and/or BAL levels of BDG of HIV/PCP-positive patients were higher than HIV-positive PCP-negatives (SMD: 2.664; 95% CI:1.110 to 4.217; $I^2$: 97.7%; P = 0.000)(Table 2).

**Table 3. The intercontinental distribution of HIV-associated PCP.**

| Continent | Asia | Europe | Africa | America |
|---|---|---|---|---|
| Prevalence | 52% | 43% | 24% | 13% |
|  | (95% CI: 32 to 73) | (95% CI: 22 to 64) | (95% CI: 12 to 35) | (95% CI: 10 to 17) |

### Risk factors for PCP

The random effect model for OR analysis indicated that smokers [56, 64, 76] (OR: 2.25; 95% CI 0.578 to 8.753) and patients with tuberculosis [39, 48, 76] (OR: 2.15; 95% CI 0.311 to 14.879) had a higher chance for PCP.

## Discussion

Before the beginning of the HIV/AIDS epidemic in the 1980s, PCP was uncommon. But soon became one of the major AIDS-defining illnesses in the world. Also, during the COVID-19 pandemic, PCP, known as a COVID mimic, caused major health problems alongside superinfection [93, 94]. Liu *et al.* performed two valuable studies about the host immune responses to COVID superinfection [95, 96]. These studies indicated that immunity triggered by SARS-CoV-2 may restrict the pathogenicity of the second post-COVID infection, such as PCP. Also, several cytokine biomarkers may be the same in these two infections, alongside the clinical sign & symptoms. In the late 1980s, 75% of people living with AIDS developed PCP [97]. A study estimated the prevalence of HIV/AIDS in the U.S. and Canada, PCP was the most common opportunistic infection during 2008–2010 [98]. There were 10,590 hospitalizations due to PCP in the U.S. in 2017 [99]. Moreover, PCP is also a common opportunistic infection among people living with HIV/AIDS in developing countries [99]. Although the prevalence of HAP has significantly decreased due to ART and prophylactic therapy with TMP/SMX, it remains a public health concern [98, 100].

In this study, a total of 38,554 HIV/AIDS patients were included, and PCP was found in 2880 of them. However, 4,044 PCP cases were reported among 79,893 HIV-negative patients (Table 2). We resulted a pooled prevalence rate of 35.4% (95% CI: 23.8 to 47.9) for HAP and 10.16% (95% CI: 2 to 25.3) for PCP among the HIV-negative population(Table 2). Also, HIV-positive patients were susceptible to PCP 12 times more than the HIV-negative population (OR: 11.710;95% CI:5.420 to 25.297)(Table 2; Fig 2; S1 Fig). Here, we indicated the worldwide distribution of the HAP; patients were reported from four continents and distributed in eighteen countries(Tables 1 and 2). Asia had the highest prevalence rate (52%), followed by Europe (43%), Africa (24%), and America had the lowest prevalence rate, 13%. The prevalence percent rate by country indicated that Japan is the highest (97.79%; 443/453), followed by Taiwan (82.35%; 14/17) and Thailand (64.66%; 97/150). The lowest rate was reported for Tanzania (0.47%; 1/213). However, in HIV-negative population prevalence percent rate of PCP was highest in Taiwan 86.29% (107/124), followed by Japan 84.73% (3631/4285), Cameroon 18.9% (21/111), and India 16.59% (37/223). These rates were zero for USA 0% (0/70418), Zambia 0% (0/1115), and Tanzania 0% (0/279).

Unfortunately, we did not find a meta-analysis study that reports the global prevalence of HAP to compare with our results. However, two meta-analysis studies investigated the prevalence of PCP among HIV-positive adults in Africa [101, 102]. Wasserman *et al.* [101] studied 6,884 individuals from 18 countries in sub-Saharan Africa. The pooled prevalence of PCP among 6,018 patients from all clinical settings was 15.4% (95% CI: 12.9 to 18.0) and was highest amongst inpatients, 22.4% (95% CI: 17.2 to 27.7). They resulted that there was a trend of

decreasing PCP prevalence amongst inpatients over time, from 28% (95% CI: 21 to 34) in the 1990s to 9% (95% CI: 8 to 10) after 2005. Another study by Wills *et al.* [102] conducted a systematic review and meta-analysis to evaluate *P. jirovecii* prevalence in African HIV-positive adults with or without respiratory symptoms. They reported a prevalence rate of 19% (95% CI: 12 to 27) among 3,583 symptomatic and a 9% [95% CI: 0 to 45] prevalence rate among 140 asymptomatic adults. They concluded that despite increased access to local HIV services, *P. jirovecii* remains common in Africa.

We resulted that the mortality rate increased by 52% in HAP patients compared to the HIV-positive PCP-negative population (OR: 1.522; 95% CI:0.959 to 2.416) (Table 2). Wasserman *et al.* [101] resulted an 18.8% (95% CI: 11.0 to 26.5)case fatality rate that PCP accounted for 6.5% (95% CI: 3.7 to 9.3) of their study deaths. Sonego *et al.* [103] designed a meta-analysis to evaluate risk factors for death from acute lower respiratory infections (ALRI) in 19,8359 children in low- and middle-income countries. Although they didn't target HAP patients, they indicated that diagnosis of *P. Carinii* and HIV/AIDS increased the risk of death (OR: 4.79; 95% CI: 2.67 to 8.61) and (OR: 4.68; 95% CI: 3.72 to 5.90), respectively. Also, they indicated that personal factors, such as age below two months (OR: 5.22; 95% CI:1.70 to 16.03), chronic underlying diseases (OR: 4.76; 95% CI: 3.27 to 6.93), severe malnutrition (OR: 4.27; 95% CI: 3.47 to 5.25) increased that risk of death. Moreover, socioeconomic and environmental factors, such as maternal age (OR:1.84; 95% CI: 1.03 to 3.31); low maternal education (OR: 1.43; 95% CI: 1.13 to 1.82); low socioeconomic status (OR: 1.62; 1.32 to 2.00), indoor air pollution (OR: 3.02; 95% CI: 2.11 to 4.31) are significantly associated with increased death rates. However, immunization (OR: 0.46; 95% CI: 0.36 to 0.58) and good antenatal practices (OR: 0.50; 0.31 to 0.81) were associated with decreased odds of death. National-scale strategies seem needed to reduce mortality rates mediated by pulmonary infections, chiefly PCP, among HIV-positive populations.

We resulted that HIV/PCP-positive patients received PCP prophylaxis six times more than HIV-positive PCP-negative cases (Table 2). The main prophylactic agent in our analyzed studies was TMP/SMX. Although there was a lack of meta-analysis studies about the status of prophylactic agents' consumption in HAP patients [104], we found three meta-analysis [105–107] of adjunctive corticosteroid therapy in these patients; however, we excluded one [105] due to the same authors' list and published study. Bucher *et al.* [104] reported the suitable outcomes of TMP/SMX on 1,448 HIV/PCP-positive patients and indicated that TMP/SMX decreased the risk ratio (RR) of PCP (versus aerosolized pentamidine: RR:0.59; 95% CI: 0.45 to 0.76&versus dapsone/pyrimethamine: RR:0.4995% CI: 0.26 to 0.92). Briel *et al.* [106] resulted that RR for overall mortality for adjunctive corticosteroids was 0.56 (95% CI: 0.32 to 0.98) at one month and 0.68 (95% CI: 0.50 to 0.94) at 3 to 4 months of follow-up. In 2015 Ewald *et al.* [107], which have the same authors list with Briel *et al.*, indicated that RR for overall mortality for adjunctive corticosteroids were 0.56 (95% CI: 0.32 to 0.98) at one month and 0.59 (95% CI: 0.41 to 0.85) at three to four months of follow-up. However, the number and size of trials of two recent studies were small (six eligible studies). Also, we found that ART in HIV/PCP-positive patients was used three times more than in HIV-positive PCP-negative patients (OR: 3.356; 95% CI:0.785 to 14.349)(Table 2). Bucher *et al.* [104] reported that 80.5% of their studied patients had received ART.

Here we indicated that HIV-positive men had a 7% higher chance rate for PCP than women (OR:1.073; 95% CI:0.674 to 1.706)(Table 2). Therefore, gender is not a potent risk factor for PCP among HIV-positive patients. However, Wasserman *et al.* [101] indicated that 39.6% of their study population were men. Also, SMD analysis of 1,583 PCP patients among 2,044 HIV-positive cases revealed no significant difference between HIV/AIDS patients with PCP and without PCP (SMD: -0.140; 95% CI:-0.315 to -0.034). Our SMD analysis showed that

BDG serum mean levels in HIV/PCP patients were higher than in HIV-positive PCP-negative cases. Despite CD4$^+$ T cells and LDH, no difference was observed in mean serum levels among the two groups. Bucher *et al.* [104] found that their studied patients' mean CD4$^+$ T cell counts ranged from 51 to 200 cells/mm3 (group mean, 115 cells/mm3). We found that HIV-positive smokers and tuberculosis patients are twice more susceptible to PCP. De *et al.* [108] investigated the influence of smoking cessation on PCP incidence in HIV patients. They indicated no evidence to support that smoking increased the incidence of PCP. However, they reported that current smokers were at higher risk of bacterial pneumonia than current non-smokers (HR: 1.73; 95%CI: 1.44 to 2.06).

During a valuable study, Elango *et al.* [109] investigated the epidemiology of PCP among HIV-positive patients in the U.S. Due to the publication bias and heterogeneity problems, we could not include this study in our final analysis; however, we read carefully and analyzed the OR of PCP-associated risk factors that were assayed in this study. They studied 148,624 HAP and 2,863,099 HIV-positive PCP-negative patients. The most associated risk factors for PCP are as follows; peripheral vascular diseases (OR: 3.467; 95% CI: 3.217 to 3.737), metastatic cancer (OR: 3.129; 95% CI: 2.865 to 3.417), paralysis (OR: 2.533; 95% CI: 2.387 to 2.687), and followed by solid tumor without metastasis (OR: 2.209; 95% CI: 2.076 to 2.350), rheumatoid arthritis (OR: 1.985; 95% CI: 1.816 to 2.165), diabetes mellitus (OR: 1.904; 95% CI: 1.866 to 1.943), Obesity (OR: 1.854; 95% CI: 1.785 to 1.926), renal failure (OR: 1.845; 95% CI: 1.816 to 1.893), hypertension (OR: 1.822; 95% CI: 1.797 to 1.846), chronic blood loss anemia (OR: 1.814; 95% CI: 1.705 to 1.929), hypothyroidism (OR: 1.716; 95% CI: 1.647 to 1.787), neurological disorders (OR: 1.552; 95% CI:1.516 to 1.589), chronic liver diseases (OR: 1.529; 95% CI: 1.500 to 1.559), lymphoma (OR: 1.531; 95% CI: 1.470 to 1.594), psychosis (OR: 1.502; 95% CI: 1.469 to 1.537), depression (OR: 1.360; 95% CI: 1.334 to 1.386), drug abuse (OR: 1.249; 95% CI: 1.232 to 1.266), valvular diseases (OR: 1.201; 95% CI: 1.149 to 1.257), congestive heart failure (OR: 1.147; 95% CI: 1.117 to 1.177). Moreover, peptic ulcer diseases (OR: 1.030; 95% CI: 0.854 to 1.241) and coagulopathy (OR: 0.967; 95% CI: 0.949 to 0.986) have not differed between PCP- positive and negatives. Moreover, the following underlying diseases are not considered a risk factor for PCP among HIV patients: chronic lung diseases (OR: 0.052; 95% CI: 0.051 to 0.053), weight loss (OR: 0.411; 95% CI: 0.406 to 0.417), fluid and electrolyte disorders (OR: 0.616; 0.610 to 0.622), deficiency anemias (OR: 0.668; 0.661 to 0.675), and pulmonary circulation disorders (OR: 0.681; 0.655 to 0.707).

## Conclusion

Now, PCP is still a serious health concern for people living with HIV/AIDS or other conditions with a weakened immune system. The exact number of HAP cases is challenging to estimate because there is no national surveillance program in different countries. We reported a high prevalence, risk, and mortality rates of PCP among the HIV-positive population. We concluded that, despite ART and prophylactic therapy, prevalence and mortality rates of HAP remained high. Therefore, the control and management strategies should be revised and updated by health policy-makers on a worldwide scale. For better management and understanding of the epidemiology and characteristics of this coinfection, designing a large-scale meta-analysis is recommended every three years.

## Supporting information

**S1 Checklist. PRISMA guideline.**
(DOCX)

**S1 Fig. L'Abbe and funnel plots for prevalence OR of HAP patients.**
(DOCX)

## Acknowledgments

The authors are grateful to the Departments of Medical Mycology and Parasitology, Schools of Medicine, Tabriz and Shiraz Universities of Medical Sciences; also, Infectious and Tropical Disease Research Center, Tabriz University of Medical Sciences for their excellent supports.

## Author Contributions

**Conceptualization:** Ehsan Ahmadpour, Hamid Morovati.

**Data curation:** Ehsan Ahmadpour, Sevda Valilou, Mohammad Ali Ghanizadegan, Rouhollah Seyfi, Seyed Abdollah Hosseini, Kareem Hatam-Nahavandi, Hanieh Hosseini, Mahsa Behravan, Hamid Morovati.

**Formal analysis:** Ehsan Ahmadpour, Sevda Valilou, Mohammad Ali Ghanizadegan, Rouhollah Seyfi, Seyed Abdollah Hosseini, Kareem Hatam-Nahavandi, Hanieh Hosseini, Mahsa Behravan, Aleksandra Barac, Hamid Morovati.

**Investigation:** Ehsan Ahmadpour, Sevda Valilou, Mohammad Ali Ghanizadegan, Rouhollah Seyfi, Seyed Abdollah Hosseini, Kareem Hatam-Nahavandi, Aleksandra Barac, Hamid Morovati.

**Methodology:** Ehsan Ahmadpour, Hamid Morovati.

**Project administration:** Hamid Morovati.

**Resources:** Hamid Morovati.

**Software:** Ehsan Ahmadpour, Hamid Morovati.

**Supervision:** Hamid Morovati.

**Validation:** Ehsan Ahmadpour, Hamid Morovati.

**Visualization:** Ehsan Ahmadpour, Hamid Morovati.

**Writing – original draft:** Hanieh Hosseini, Mahsa Behravan, Hamid Morovati.

**Writing – review & editing:** Ehsan Ahmadpour, Aleksandra Barac, Hamid Morovati.

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
