## [Decision Letter · Decision Letter 0]

10 Dec 2023

PONE-D-23-39809Global Prevalence, Mortality, and Main Characteristics of HIV-associated Pneumocystosis: A Systematic Review and Meta-AnalysisPLOS ONE

Dear Dr. Morovati,

Thank you for submitting your manuscript to PLOS ONE. After careful consideration, we feel that it has merit but does not fully meet PLOS ONE’s publication criteria as it currently stands. Therefore, we invite you to submit a revised version of the manuscript that addresses the points raised during the review process.

We look forward to receiving your revised manuscript.

Kind regards,

Benjamin M. Liu, MBBS, PhD, D(ABMM), MB(ASCP)

Academic Editor

PLOS ONE

Journal Requirements:

Burden of pneumocystis pneumonia in HIV-infected adults in sub-Saharan Africa: a systematic review and meta-analysis - https://doi.org/10.1186/s12879-016-1809-3

Adjunctive corticosteroids for Pneumocystis jiroveci pneumonia in patients with HIV infection (Review) - https://doi.org/10.7892/boris.68082

Pneumocystis pneumonia - https://www.cdc.gov/fungal/diseases/pneumocystis-pneumonia/

In your revision ensure you cite all your sources (including your own works), and quote or rephrase any duplicated text outside the methods section. Further consideration is dependent on these concerns being addressed.

4. We note that Figure 3 in your submission contain map/satellite images which may be copyrighted. All PLOS content is published under the Creative Commons Attribution License (CC BY 4.0), which means that the manuscript, images, and Supporting Information files will be freely available online, and any third party is permitted to access, download, copy, distribute, and use these materials in any way, even commercially, with proper attribution. For these reasons, we cannot publish previously copyrighted maps or satellite images created using proprietary data, such as Google software (Google Maps, Street View, and Earth). For more information, see our copyright guidelines: http://journals.plos.org/plosone/s/licenses-and-copyright.

a. You may seek permission from the original copyright holder of Figure 3 to publish the content specifically under the CC BY 4.0 license.  

5. We note that this manuscript is a systematic review or meta-analysis; our author guidelines therefore require that you use PRISMA guidance to help improve reporting quality of this type of study. Please upload copies of the completed PRISMA checklist as Supporting Information with a file name “PRISMA checklist”.

Reviewers' comments:

Reviewer's Responses to Questions

**Comments to the Author**

1. Is the manuscript technically sound, and do the data support the conclusions?

Reviewer #1: Yes

2. Has the statistical analysis been performed appropriately and rigorously? 

Reviewer #1: Yes

3. Have the authors made all data underlying the findings in their manuscript fully available?

Reviewer #1: Yes

4. Is the manuscript presented in an intelligible fashion and written in standard English?

Reviewer #1: Yes

5. Review Comments to the Author

Reviewer #1: 1. In abstract, the authors stated that a large-scale meta-analysis is recommended every three years. Why does it have to every THREE years? This part should be revised or removed.

2. Please give a full term when an abbreviation appears for the first time.

3. The authors are encouraged to discuss more on the progress and challenges of PCR-based diagnosis of PCP based on the following reference, if deemed fit.

Liu B, Totten M, Nematollahi S, Datta K, Memon W, Marimuthu S, Wolf LA, Carroll KC, Zhang SX. Development and Evaluation of a Fully Automated Molecular Assay Targeting the Mitochondrial Small Subunit rRNA Gene for the Detection of Pneumocystis jirovecii in Bronchoalveolar Lavage Fluid Specimens. J Mol Diagn. 2020 Dec;22(12):1482-1493. doi: 10.1016/j.jmoldx.2020.10.003. Epub 2020 Oct 15. Erratum in: J Mol Diagn. 2021 Apr;23(4):506. PMID: 33069878.

4. "Without treatment, PCP can cause death [8]." This statement should be modified. Besides death, what are other outcomes of poor care of PCP?

5. The authors stated that PCP is known as a COVID mimic, caused major health problems alongside superinfection. A brief discussion on the difference/differentiation between PCP and COVID, e.g., cytokine biomarkers and virus detection, will be needed to avoid confusion to the readers. Some references are suggested as shown below.

Clinical significance of measuring serum cytokine levels as inflammatory biomarkers in adult and pediatric COVID-19 cases: A review. Cytokine. 2021 Jun;142:155478. doi: 10.1016/j.cyto.2021.155478. Epub 2021 Feb 23. PMID: 33667962; PMCID: PMC7901304.

Role of Host Immune and Inflammatory Responses in COVID-19 Cases with Underlying Primary Immunodeficiency: A Review. J Interferon Cytokine Res. 2020 Dec;40(12):549-554. doi: 10.1089/jir.2020.0210. PMID: 33337932; PMCID: PMC7757688.

6. PLOS authors have the option to publish the peer review history of their article (what does this mean?). If published, this will include your full peer review and any attached files.

Reviewer #1: No

---

## [Author Response · Author response to Decision Letter 0]

8 Jan 2024

January 8, 2024

Dear Professor Benjamin M. Liu 

Academic Editor of PLOS ONE

We are privileged to submit the revised version of manuscript ID ‘PONE-D-23-39809’ entitled: “Global Prevalence, Mortality, and Main Characteristics of HIV-associated Pneumocystosis: A Systematic Review and Meta-Analysis" for your consideration for publication in the PLOS ONE. We would like to thank you and the Reviewers for their constructive comments that have enabled us to improve the quality of the manuscript. Please find our responses to the comments/questions of the Reviewers/Editorial board point by point as follows. We indicated the changes in the manuscript with tracked changes (by page and line number).

Thank you for your time and consideration of this manuscript. Please do not hesitate to contact us should additional information be required.

Sincerely yours

*Corresponding author: Dr. Hamid Morovati 

Department of Parasitology and Mycology, School of Medicine, Shiraz University of Medical Sciences, Shiraz, Iran. Email: morovatihamid1989@gmail.com; morovati@sums.ac.ir

Address: Third floor, Building 1, School of Medicine, Zand Street, Imam Hossein Square, Shiraz, Iran.

Tel & Fax: +98 713 230 4982

 

Journal Requirements:

If applicable, we recommend that you deposit your laboratory protocols in protocols.io to enhance the reproducibility of your results. Protocols.io assigns your protocol its own identifier (DOI) so that it can be cited independently in the future. 

For instructions see: https://journals.plos.org/plosone/s/submission-guidelines#loc-laboratory-protocols. 

Additionally, PLOS ONE offers an option for publishing peer-reviewed Lab Protocol articles, which describe protocols hosted on protocols.io. Read more information on sharing protocols at https://plos.org/protocols?utm_medium=editorial-email&utm_source=authorletters&utm_campaign=protocols.

We look forward to receiving your revised manuscript.

We thank your positive feedback and constructive comments on the manuscript.

Query 1- Please ensure that your manuscript meets PLOS ONE's style requirements, including those for file naming. The PLOS ONE style templates can be found at 

Response: Special thanks for valuable information. We checked the links and revised our manuscript according to the journal guidelines.

Query 2- We noticed you have some minor occurrence of overlapping text with the following previous publication(s), which needs to be addressed: 

Burden of pneumocystis pneumonia in HIV-infected adults in sub-Saharan Africa: a systematic review and meta-analysis - https://doi.org/10.1186/s12879-016-1809-3

Adjunctive corticosteroids for Pneumocystis jiroveci pneumonia in patients with HIV infection (Review) - https://doi.org/10.7892/boris.68082

Pneumocystis pneumonia - https://www.cdc.gov/fungal/diseases/pneumocystis-pneumonia/

In your revision ensure you cite all your sources (including your own works), and quote or rephrase any duplicated text outside the methods section. Further consideration is dependent on these concerns being addressed.

Response: Thank you. We rechecked these links according to your recommendation. These studies previously have been discussed in our manuscript for several times. The study by Ewald et al. was discussed and cited in the discussion section (line 296). The study by Wasserman et al. was discussed in several places of the discussion section; such as line 258, 270, and 305.

Query 3- We note that you have stated that you will provide repository information for your data at acceptance. Should your manuscript be accepted for publication, we will hold it until you provide the relevant accession numbers or DOIs necessary to access your data. If you wish to make changes to your Data Availability statement, please describe these changes in your cover letter and we will update your Data Availability statement to reflect the information you provide.

Response: Thanks. We checked our submission proof. Repository information statement was “all relevant data are within the manuscript and its supporting information files” and data availability statements was “All data are fully available without restriction”. There is no repository information of data at acceptance. We declare these statements in the cover letter.

Query 4- We note that Figure 3 in your submission contain map/satellite images which may be copyrighted. All PLOS content is published under the Creative Commons Attribution License (CC BY 4.0), which means that the manuscript, images, and Supporting Information files will be freely available online, and any third party is permitted to access, download, copy, distribute, and use these materials in any way, even commercially, with proper attribution. For these reasons, we cannot publish previously copyrighted maps or satellite images created using proprietary data, such as Google software (Google Maps, Street View, and Earth). For more information, see our copyright guidelines: http://journals.plos.org/plosone/s/licenses-and-copyright.

Response: Thank you. We decided to remove the figure three and replaced it with the table 3.

Query 5- We note that this manuscript is a systematic review or meta-analysis; our author guidelines therefore require that you use PRISMA guidance to help improve reporting quality of this type of study. Please upload copies of the completed PRISMA checklist as Supporting Information with a file name “PRISMA checklist”.

Response: Thank you. We renamed the file name to PRISMA checklist and listed as the supporting information file. 

Reviewers' comments:

Reviewer #1:

Query 6- In abstract, the authors stated that a large-scale meta-analysis is recommended every three years. Why does it have to every THREE years? This part should be revised or removed.

Response: Special thanks for your great comment. Following your recommendation, we decided to remove the part which you have concerned. The changes were indicated with the track change. 

Query 7- Please give a full term when an abbreviation appears for the first time.

Response: Thank you. We carefully revised the manuscript for any writing errors. All errors were fixed. However, we didn’t find abbreviation missuses. If it possible, kindly address your concerned error.

Query 8- The authors are encouraged to discuss more on the progress and challenges of PCR-based diagnosis of PCP based on the following reference, if deemed fit.

Liu B, Totten M, Nematollahi S, Datta K, Memon W, Marimuthu S, Wolf LA, Carroll KC, Zhang SX. Development and Evaluation of a Fully Automated Molecular Assay Targeting the Mitochondrial Small Subunit rRNA Gene for the Detection of Pneumocystis jirovecii in Bronchoalveolar Lavage Fluid Specimens. J Mol Diagn. 2020 Dec;22(12):1482-1493. doi: 10.1016/j.jmoldx.2020.10.003. Epub 2020 Oct 15. Erratum in: J Mol Diagn. 2021 Apr;23(4):506. PMID: 33069878.

Response: Thanks for the comment. We carefully read this valuable study and apply that in our manuscript (lines 66 to 68).

Query 9- "Without treatment, PCP can cause death [8]." This statement should be modified. Besides death, what are other outcomes of poor care of PCP?

Response: Thanks for the comment. We removed and replaced the concerned term as you recommended.

Query 10- The authors stated that PCP is known as a COVID mimic, caused major health problems alongside superinfection. A brief discussion on the difference/differentiation between PCP and COVID, e.g., cytokine biomarkers and virus detection, will be needed to avoid confusion to the readers. Some references are suggested as shown below.

Clinical significance of measuring serum cytokine levels as inflammatory biomarkers in adult and pediatric COVID-19 cases: A review. Cytokine. 2021 Jun;142:155478. doi: 10.1016/j.cyto.2021.155478. Epub 2021 Feb 23. PMID: 33667962; PMCID: PMC7901304.

Role of Host Immune and Inflammatory Responses in COVID-19 Cases with Underlying Primary Immunodeficiency: A Review. J Interferon Cytokine Res. 2020 Dec;40(12):549-554. doi: 10.1089/jir.2020.0210. PMID: 33337932; PMCID: PMC7757688.

Response: Thanks for the great suggestion. We carefully read these valuable studies and apply them in our manuscript (lines 244 to 247).

---

## [Editor Report · Decision Letter 1]

10 Jan 2024

Global Prevalence, Mortality, and Main Characteristics of HIV-associated Pneumocystosis: A Systematic Review and Meta-Analysis

PONE-D-23-39809R1

Dear Dr. Morovati,

We’re pleased to inform you that your manuscript has been judged scientifically suitable for publication and will be formally accepted for publication once it meets all outstanding technical requirements.

Kind regards,

Benjamin M. Liu, MBBS, PhD, D(ABMM), MB(ASCP)

Academic Editor

PLOS ONE
---

## [Editor Report · Acceptance letter]

8 Feb 2024

PONE-D-23-39809R1 

PLOS ONE

Dear Dr. Morovati, 

I'm pleased to inform you that your manuscript has been deemed suitable for publication in PLOS ONE. Congratulations! Your manuscript is now being handed over to our production team.

Kind regards, 

on behalf of

Dr. Benjamin M. Liu 

Academic Editor

PLOS ONE